:ᐧPLOS | ONE

# Agronomic or contentious land change? A longitudinal analysis from the Eastern Brazilian Amazon

Stephen P. Aldrich[1]*, Cynthia S. Simmons[2], Eugenio Arima[3], Robert T. Walker[4], Fernando Michelotti[5], Edna Castro[6]

**1** Department of Earth and Environmental Systems, Indiana State University, Terre Haute, IN, United States of America, **2** Department of Geography, University of Florida, Gainesville, FL, United States of America, **3** Department of Geography and The Environment, University of Texas-Austin, Austin, TX, United States of America, **4** Center for Latin American Studies and Department of Geography, University of Florida, Gainesville, Florida, United States of America, **5** Institute for Agrarian and Regional Development Studies, Federal University of the South and Southeast of Pará Marabá, Pará Marabá, Pará, Brazil, **6** Center for Amazon Studies, Federal University of Pará, Belém, Pará, Brazil

\* steve.aldrich@indstate.edu

**Data Availability Statement:** All relevant data are within the paper and its Supporting Information files. These data are also available from Sycamore Scholars, the institutional repository for Indiana

## Abstract

Since 1984, nearly 1,000 people have been killed in the Brazilian Amazon due to land conflicts stemming from unequal distribution of land, land tenure insecurity, and lawlessness. During this same period, the region experienced almost complete deforestation (< 8% forest cover by 2010). Land conflict exacts a human toll, but it also affects agents' decisions about land use, the subject of this article. Using a property-level panel dataset covering the period of redemocratization in Brazil (1984) until the privatization of long-term leases in the Eastern Amazon (2010), we show that deforestation is affected by land conflict, particularly in cases of expropriation of property for agrarian reform settlement formation and when that conflict involves fatalities. Deforestation on agrarian reform settlements is much greater when soils are poor for agriculture and when the land has been the object of past conflict. Deforestation and conflict are episodic, and both agronomic drivers and contentious drivers of land change are active in the region. Ultimately, the outcome of these processes of contentious and agronomic land change is substantial deforestation, regardless of who was in possession and control of the land.

## 1) Introduction

Conflict over access to natural resources and land have emerged as important drivers of land change [1–4]. Land is the most important economic asset and sole source of income or subsistence for millions of families [5–7]. In many parts of the world where socio-economic exclusion and marginalization lead to unequal access to land, conflicts emerge resulting in environmental degradation in many cases [8–11]. The nexus between land conflict and threats to natural resources is a widespread phenomenon. Transnational social movements like *La Via Campesina* have recently stimulated the rural poor to question access to land, often limited

State University, at http://scholars.indstate.edu/handle/10484/12380?show=full.

**Funding:** Aldrich, Simmons, Arima: National Science Foundation, BCS-1157521. https://nsf.gov/awardsearch/showAward?AWD_ID=1157521. The funders had no role in study design, data collection and analysis, decision to publish, or preparation of the manuscript.

**Competing interests:** The authors have declared that no competing interests exist.

for institutional and historical reasons at odds with social equity. These movements have motivated a rural constituency that aggressively seeks more dignified, farm-based livelihoods. As a consequence, contention over land has seized headlines throughout the Global South, with notable movements in South America including ongoing conflict and land reform in Bolivia [12, 13], Paraguay [14], and Brazil [2, 15–18]. Further afield, notable land reform movements also have transformed land tenure discussions in Southeast Asia in the Philippines [19, 20] and Indonesia [1, 21], and in Africa [22], including Liberia [23, 24], Zimbabwe [25–27], and South Africa [28–30]. We speculate that when such social phenomena manifest in forest-rich nations, deforestation occurs for reasons of social conflict in addition to more traditional agronomic pressure, and therefore these nations require policy interventions tuned to the distal factors that are sustaining social inequalities.

The Brazilian Amazon is no exception to this narrative. Over the past three decades conflict over land in the Amazon has claimed the lives of 995 people [31], with well over half of Amazonian municipalities experiencing at least one land conflict since 1985. Previous research has identified many factors driving Amazonian deforestation including fiscal incentives, transportation costs, migration, and household processes. The link between land conflict and deforestation has also been investigated recently [32, 33–35], including in other regions [36, 37]. Between 1964 and 1997 land reform pursuant to such conflict may have accounted for 30 percent of Amazonian deforestation, overall [38]. Despite Brazil's prohibition for new land reform settlements in forested areas, this is where the vast majority are formed [32, 33, 38, 39]. As Brown et al. [34] demonstrate, deforestation is greater on settlements formed by means of direct action land reform (DALR) occupations, and in municipalities adjacent to conflict areas. Other recent work has also shown that agricultural markets and speculation also significantly enhance deforestation rates [e.g., 35], raising the question whether both processes are at work in the Amazonian landscape. The study presented here shows a direct, and more nuanced, link between deforestation at the property-level and contentious interactions between large landholders and the landless in their struggle to claim properties over a 27 year period. Specifically, we analyze the effects of back and forth changes in the control of land between those two groups on land use decisions and long-term effects on deforestation.

The objective of the article is therefore to expand the explanatory repertoire of Land Change Science (LCS) by assessing the impact of land conflict on deforestation, referred to here as Contentious Land Change (CLC). To achieve this objective, the article focuses on the destruction of the Brazil Nut forest (~20,000 Km$^2$) in Southeastern Para State, the state which accounts for nearly 17 percent of all Amazonian deforestation (Fig 1; INPE 2011). Until recently the Brazil Nut Polygon (BNP) formed unusually dense concentrations of Brazil Nut trees (*Bertholletia excelsa*, *Lecythidaceae*) that sustained a profitable extractive economy in the first half of the 20th century. But the BNP lay in the development pathway of Brazil's Military Government (1964–1985), for it was here that road-building crews opened the Amazonian forest to occupation with the construction of BR-230, the so-called Transamazon Highway. Government incentives attracted capital from the south and landless farmers from the northeast, disrupting land claims of the Brazil Nut oligarchs and setting in motion an interwoven process of social conflict and environmental destruction that continues to this day [32, 38, 40–43]. The region is now almost completely deforested (<8% forest cover remains), and fights for land there continue to claim lives [32, 44, 45].

We pursue our objectives as follows. In the next section, we provide a background discussion of the study region, its development and integration with the Brazilian economy, and history of conflict. Next, we articulate a theoretical model of CLC that distinguishes land cover change resulting from agronomic decision-making and change driven by land conflict, from which we derive the primary research hypotheses that land conflict increases deforestation.

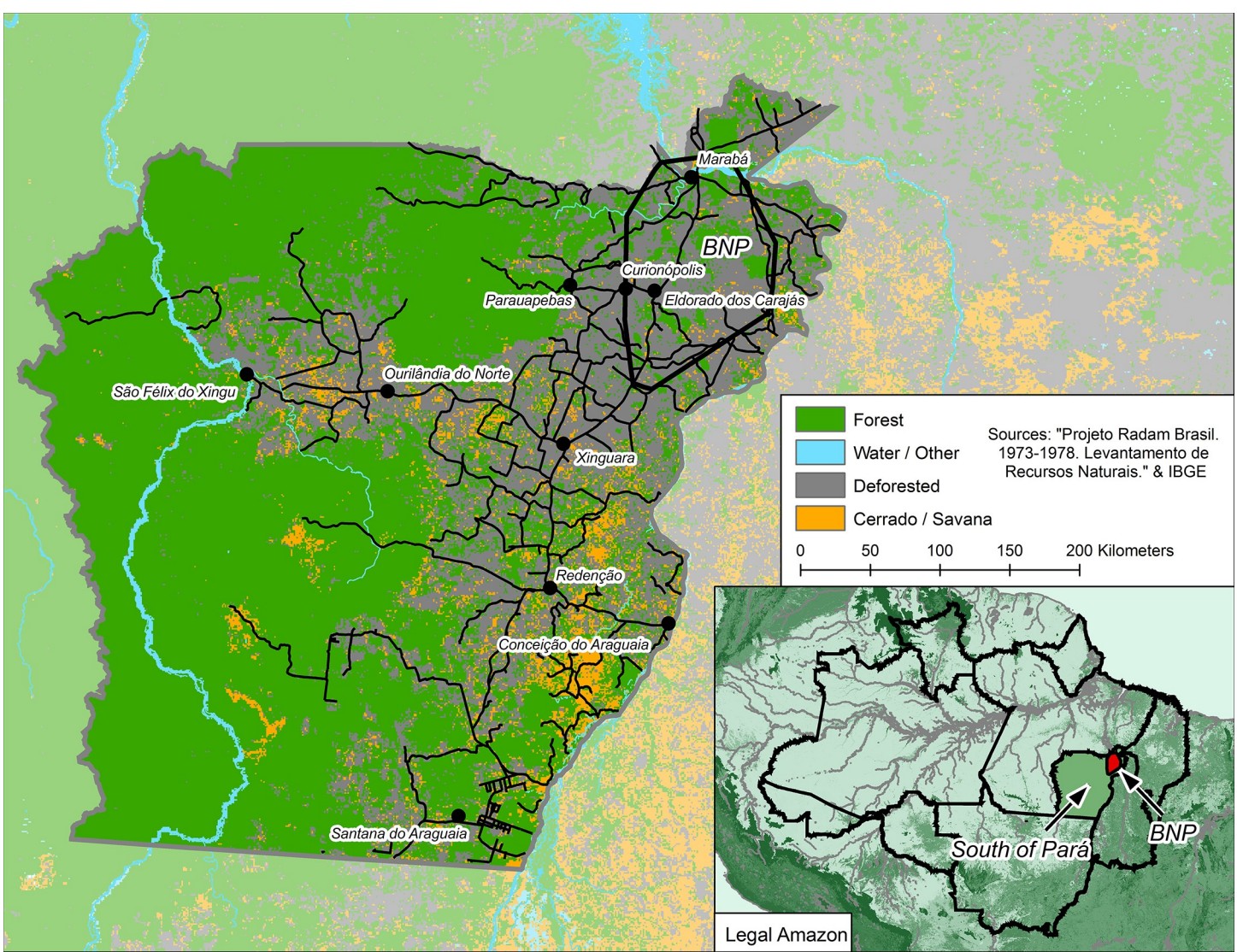

**Fig 1. Dominant vegetation pre-1970 in Southeastern Pará, Brazil.** The Brazil Nut Polygon is in the upper right (Northeast) corner of the region. Southeastern Pará straddles the ecotone between closed-canopy forests of the Amazon and the Savanah environments of Brazil's Cerrado.

Regression analyses, using panel data enabled by a unique set of geographic and historical data covering the period of redemocratization (1984–1985) through formal land regularization in 2010 [46], are employed in order to explicate CLC and empirically assess the significance of land conflict on deforestation. These include (1) a multi-decadal satellite image series covering 27 years (1984–2010); (2) cadastral maps for large holdings (4,886 ha average) and agrarian reform settlements; and (3) a newspaper archive detailing land conflicts unfolding at property scale for the same period as the satellite imagery (more than 8,000 individual pages). This information enables the systematic evaluation of deforestation associated with the social inequalities driving land conflict in the Amazon Basin. Thus, the article provides evidence for an important driver of land change, typically overlooked in deforestation studies but of likely significance given climate change, rural poverty, and land grabs occurring world-wide.

## 2) Background

Our analysis of CLC addresses Southeastern Pará, a 212,375 km$^2$ geographic expanse below the Transamazon Highway and east of the Xingu River in the state of Pará, where land owner-ship has long been in contention (Fig 1). Most of the region's land cover in the mid-1970s was dominated by closed-canopy tropical forest, comprising a variety of valuable hardwoods including mahogany (*swietenia macrophylla*), rubber trees (*hevea brasiliensis*), and Brazil nut (*bertholletia excelsa*). Early in the opening of this region to development and natural integra-tion (1964–1975) cerrado grasslands (open herbaceous land cover interspersed with shrub for-est) covered an estimated 11 percent of this region, whereas agriculture occupied most of the remainder (8 percent), primarily in the vicinity of the region's principal city, Marabá, and along roads to the South. The remaining 81 percent of the region was covered with primary forest.

Although much of the region experienced land conflict through the later part of the 20[th] century, our analytical modeling resides in its northern reaches in the BNP, an area of more than 6,800 km$^2$ located roughly between the Tocantins River and the Carajás mines (Fig 1). The BNP, much of it once covered by dense stands of Brazil Nut trees and dominated by non-timber forest extraction, reveals a notorious history of violent land conflict [2, 32, 47]. The question this research aims to address is the extent to which conflict was a factor in the forest's demise, which today amounts to a very limited fraction of its original extent.

Like most of Pará, the BNP remained isolated until the later part of the 20[th] century, although Brazil Nut extraction from concentrations of the trees, referred to as *castanhais*, stim-ulated an early boom. The *castanhais* first functioned as common properties accessible to all, but the state government of Pará enclosed most of them with long-term leases, or *aforamentos*, granted to wealthy locals. The enclosure process unfolded during the first half of the 20[th] cen-tury, creating an impoverished rural population whose ranks grew with the implementation of Brazil's ambitious development plans. The construction of a highway network (i.e., the Belem-Brasilia in 1956, PA-150 in 1964, running east-west through the BNP, and the Transamazon in 1970, running east-west through the BNP) triggered the in-migration of peasant farmers eager to escape drought and poverty in northeastern Brazil and joblessness in the South. Gold dis-coveries also sparked waves of wildcat miners. Significant numbers of the new rural poor sought to alleviate poverty by occupying small parcels on lands that largeholders had enclosed as their privatized *castanhais*, an action justified by squatter rights (*usocapião*) as afforded in the 1964 Land Statute [48, see especially Article 13], and subsequently permitted in the 1988 Constitution of Brazil [49, see articles 184–191]. The vast forested land of the BNP was the ideal target of the landless since forests were deemed unproductive and the properties too large to surveil for squatters.

Initial conflicts in the Brazil Nut region of Southeastern Pará intensified with the arrival of armed Maoist guerrillas in the late 1960s, whose range of operation extended from São Felix do Xingu in Pará, east to Estreito in Tocantins State, and from Marabá to just south of Redenção, Pará [50]. Brazil's military government declared the region a national security zone and eradicated the insurgents between 1972 and 1974 in a controversial campaign that left many casualties, and few prisoners [51, 52] and solidified the federal governments militaristic stance toward the region's landless poor. During, and immediately after (1975 onwards), this period the sons and daughters of the Brazil Nut oligarchy came home to the region after attending national and international Universities, bringing new ideas about forms of business and production to the region. Land uses in the region quickly began to shift from extensive-Brazil Nut extraction, relying on the matrix of closed-canopy forest to maintain production, to cattle ranching, which required the removal of forests. Going hand-in-hand with this shift

(which took the better part of a decade) were economic subsidies meant to encourage and "improve" a more technical cattle economy [53–55], totaling more than USD$2.3 billion in 2019 dollars. Competition for land between rich and poor continued, leading to the so-called Brazil Nut War (1984–1989), a bloody engagement pitting long-term lease (*aforamento*) holders, the Brazil Nut oligarchs and their new ranching progeny, against a peasantry in want of land [56]. The legal designation of the Brazil Nut Polygon (BNP) in the heart of the conflict zone took place at this time, as the oligarchs sought state intervention to secure their land claims, to no avail.

With the opening of popular politics following the redemocratization of Brazil in 1984–1985, land conflict dynamics shifted from relatively uncoordinated squatting by landless farmers to an organized landless movement led by social movement organizations (SMOs) dedicated to agrarian reform, specifically targeting the BNP. In Brazil, landless movements (e.g. Movement of the Landless Rural Workers, or MST) employ DALR, the most common confrontational tactic being the surprise occupation of largeholdings by 100s to 1000s of landless families. These occupations happen on properties SMOs deem vulnerable to expropriation for agrarian reform [57], as permitted by the 1988 constitution [49]. Between 1988 and 2014, more than 1.275 million families participated in DALR, and more than 40% of participating families and 70% of land occupied was in Amazonia [58]. Pará State, which contains the study area, accounts for 6.6 percent of families and nearly one-fourth of all land impacted by DALR nationwide [59, 60]. The arrival of SMOs in Amazonia in the 1990s corresponded to a marked increase in SMO-led DALR that too-often ended with disastrous consequences, as in the murder of 19 landless activists by military police on April 17, 1996, in Eldorado do Carajás [32]. All told, between 1980 and 2010, there were more than 500 land conflict related deaths throughout Southeastern Pará [61], nearly half of these (245) in the BNP. Our own data puts the land-conflict fatality count between 1980 and 2010 at over 1,000 in the BNP, suggesting watchdog groups like the CPT [31] may be undercounting violence.

Over the thirty plus years of land conflict in this region, rapid and extensive environmental change was also taking place. The BNP possessed 88% forest cover in 1984, indicating little land change up to, and including, the period of guerrilla conflict in the 1970s. As Brazil's nascent civil society allowed for more direct forms of political action, including early DALR actions, deforestation associated with agricultural expansion and land conflict intensified, and by the end of 1989 the forests of the BNP had shrunk to a third of their original extent. As DALR intensified between 1990 and 1996, deforestation continued, and the BNP's forest area decreased by another 30 percent. By 2000, 75 percent of its original extent was gone, and by 2010 less than 8 percent remained. Thus, within less than 30 years (from 1984 to 2010), the BNP experienced a nearly complete conversion from primary forest to fields and pastures. Today, the rural poor reside in agrarian reform settlements that were once BNP *castanhais* (Brazil Nut groves), and many engage in calving to supply animals to their former adversaries, the Brazil Nut oligarchs turned *fazendeiros* (ranchers), who fatten the animals and deliver them to ten modern meat-packing plants which sell chilled beef on domestic and international markets [62].

The history of this region has many global corollaries, and understanding the complex interactions between CLC and agronomic deforestation is important in the face of ongoing land conflict and agricultural expansion across the global South. Thus, the landscapes and histories of the BNP are a model landscape within which to understand how CLC and agronomic pressures interplay to result in landscape change. This extension of Land Change Science is part of the ongoing work to address the shortcomings with the concept of unitary decision-makers, and to bring social process more explicitly to Land Change Science (LCS) conceptualizations of landscape change.

## 3) Contentious land change: Conceptual framework and methods

### Conceptual framework

LCS typically focuses on (1) the proximate decision-making of local land managers, and (2) the distal and highly aggregated socio-economic environments that comprise a context for the proximal actors. This study dispenses with the LCS presumption that landholdings are managed by unitary decision-makers acting in the interest of market production. Often, deforestation results from contentious interactions between powerful elites and the rural poor, groups with differing motivations and agricultural practices. Our analysis posits an explanatory framework that acknowledges the impact of social processes, individual behaviors, and organizational actions on land change (LC), in addition to the rational economic actor, in this case a farmer or rancher. Specifically, we address environmental impacts stemming from enclosures of natural resources [4, 9, 63]. In the study area, land scarcity arose following enclosures of public lands dedicated to Brazil Nut extraction, the so-called *castanhais*. An important component of land change in forest frontiers results as land managers deforest in order to engage in agricultural activities, a process we refer to as agronomic land change, or ALC. Nevertheless, the presence of land conflict alters this dynamic by introducing new behaviors and social processes that do not conform to a static, rational actor model.

Changes in land cover in contentious settings, referred to here as contentious land change (CLC), occur over multiple decades and stem from a variety of actions taken by competing claimants, including preemptive deforestation in anticipation of seizure by alternative claimants, excessive land clearance meant to stake a claim, the use of fire as a conflict weapon, and resource mining (e.g. valuable hardwoods, soils) in advance of expected dispossession [2, 47, 64, 65]. DALR may take years to resolve, and during this time land management changes hands multiple times between original owner and participants, as DALR actions precipitate a series of land occupations starting with the initial event until the property is expropriated for the purposes of agrarian reform. Specifically, property management changes subsequent to a DALR occupation, but may revert back to the original owner if courts rule for repossession (*reintegração*) and eviction of the occupiers (*despejo*). However, frequently the property is re-occupied by DALR participants, and the process repeats, leading to a series of intervals whereby largeholders or DALR movements effectively control and undertake land change (LC) on the property in question. This process is not unique to the Amazon, and can be observed in other regions experiencing land conflict and rapid environmental transformation.

For the Amazonian case, we hypothesize that deforestation reflects both ALC and CLC, and that CLC adds to the total amount of deforestation that would otherwise occur under ALC. This is depicted in Fig 2 for two hypothetical properties of 4,000 ha, an average size for a BNP holding (BNP mean = 4,886 ha). One experiences CLC and the other does not. Both follow a linear land clearance pattern at first, a property "life cycle" [66–68]. Divergence sets in at t = 2, as DALR participants contest one of the holdings, thereby precipitating CLC. Property control then passes back and forth between DALR participants and the initial land owner, creating a series of management intervals (e.g., $t = 4 \rightarrow t = 5$, $t = 5 \rightarrow t = 6$, etc.). In the other case, the property without contention experiences ALC until 2,000 ha are cleared, or half the holding, in accordance with federal law that mandates a "Legal Reserve" of 50 percent forest [Law 7.511, Law 4.771, Law 7.803, and Medida Provisória no 2.166–67, a specification that continue to change; 47, 69].

As depicted and hypothesized, the property in contention shows more LC once DALR has initiated (at t = 2), and a higher amount of long-run, end-of-period deforestation (at t = 10). Fig 2 depicts CLC for intervals under DALR control may be "excessive," because of assumed DALR interest in establishing productive-use claims, which usually entail removing

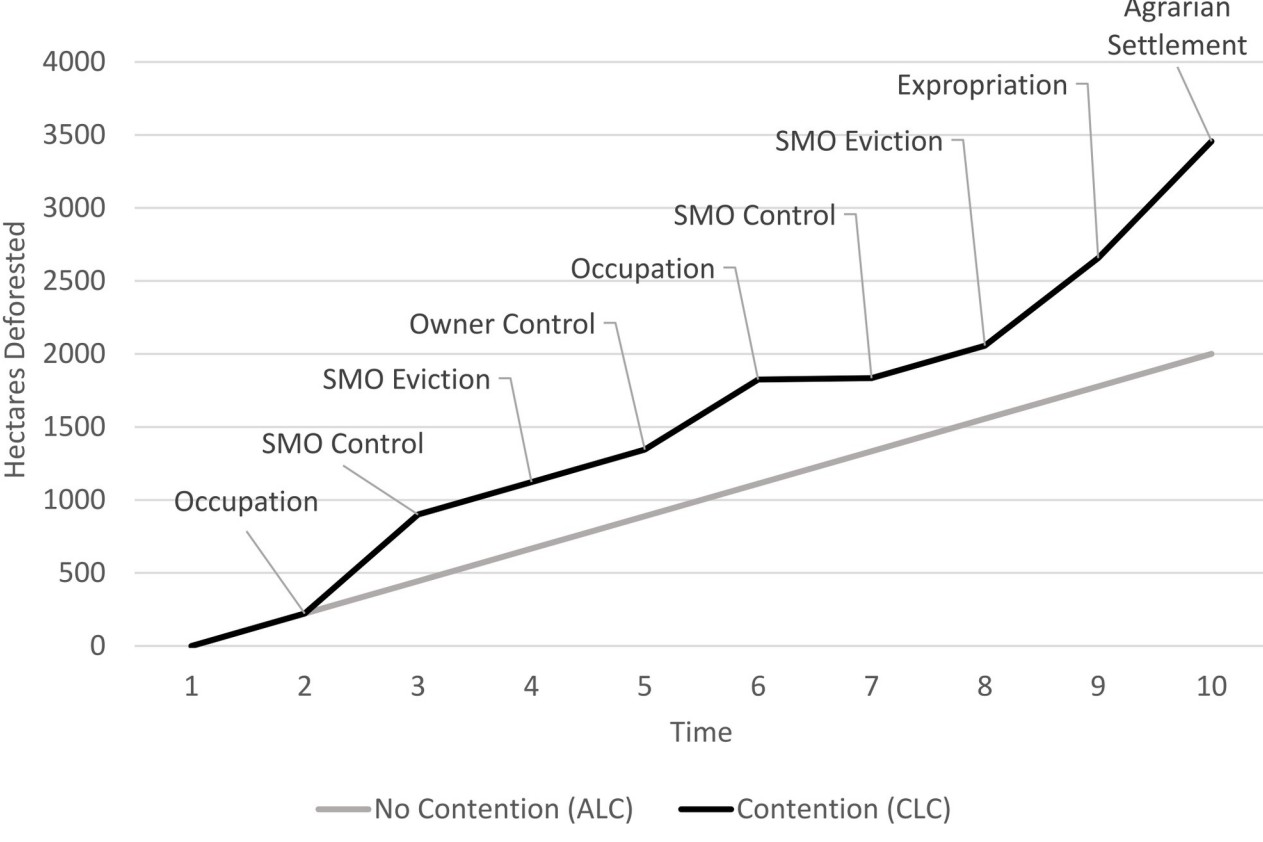

**Fig 2. Deforestation trajectories on landholdings under ALC and CLC.**

"unproductive" forests (e.g., interval t = 2 → t = 3). Initial owners show more complicated behavior. As with DALR participants, they often wish to demonstrate productive-use and are known to deforest preemptively to inhibit DALR. Alternatively, when government fails to defend their claims, there may be no incentive to improve the land and instead land owners are motivated to liquidate the value of standing forest as they await adjudication and possible expropriation. Fig 2 suggests that following expropriation of a landholding to establish a land reform settlement (t = 8), deforestation ticks up, as large numbers of DALR participants divide the property into individual lots and begin their agricultural activities.

Fig 2 presents the hypotheses tested in the article graphically, which are that land conflict augments deforestation magnitudes, such that (*H1*) deforestation is greater during those time periods with land conflict compared to periods with relative peace. We also hypothesize (*H2*) that among properties that have had land conflict, those properties that have been expropriated have greater deforestation. Although our theoretical frame elaborates different motivations for land managers to deforest during periods of contention, it does not attribute responsibility for deforestation to a specific class of land manager–the original largeholder or the DALR partici-pants–as both classes are responsible at different times. That said, LCS scholarship thus far has tended to attribute deforestation to the creation of a settlement project, thus the settlers are the culprits without much consideration of the role played by the largeholders in the DALR pro-cess. The expectation is that once a settlement is created, the cumulative impact of individual smallholders is greater than would be the situation under a unitary largeholder. The spatial-temporal dimensions of our data allow for the direct examination of whether deforestation

was greater during periods of contention or settlement formation (*H1 & H2*). Finally, in keeping with the notion of CLC as a driver of change, we hypothesize (*H3*) deforestation is greater by the end of the study period on properties that have had contention compared to those without.

## Data and methods

**3.b.i) Analytical approach.** We address our hypotheses (*H1*, *H2*, *H3*) using OLS regression, which we then extend using a series of "treatment effects" analyses in a dynamic panel environment, and pooled OLS regression. We present several models in the text, but also include other specifications in the Supporting Information files. The model takes the form

$$Y_{i,t} = \beta X_{i,t} + \alpha_i + u_{i,t}$$

Where *Y* is the dependent variable, deforestation rate or total deforestation in hectares (depending on model specification), on property *i* at time *t*, *β* is the coefficient for *X*, a binary variable indicating if conflict occurred, *α* is the intercept, and *u* is the error term. To test the first two hypothesis, we implement and compare two model variations: pooled OLS and a fixed effects Arellano-Bond (AB) panel estimation. The inclusion of a lagged dependent variable in the model is likely to bias the OLS estimation due to the correlation between $u_{i,t}$ and $Y_{i,t-1}$. The AB model produces consistent estimators by first differencing to remove the panel-level effects and by creating instruments for the lagged dependent variables within a generalized method of moments estimator framework [70]. We also estimated a third model (Arellano-Bover-Blundell-Bond, [71, 72]) that is consistent even under weaker conditions than AB. Results are not reported here because they are similar to AB, but are available upon request to the corresponding author.

The dynamic panel approach corrects the limitations of Aldrich [47] and Aldrich *et al.* [44]. In addition to the panel structure, the lagged dependent variable provides an important control for unobserved effects. It also controls for the likely time dependency (or historical path) of deforestation [e.g., 73].

For all hypotheses we included control variables (see Fig 3 and S1 Table) which captured the effect of time-variant changes in the region, such as rainfall [74], changes to the road network, and economic growth rates, etc. However, many "traditional" land cover change control variables like the quality of the resource base (e.g., soil quality) and distance to market (independent of the evolving road network) are handled automatically by our panel approach [75, 76], as is spatial autocorrelation [77: 27].

In order to control for potential bias arising from selective occupation, we adopted a two pronged approach whereby matching estimators were used as a pre-processing step to select a balanced sample of properties that are comparable with respect to certain pre-occupation characteristics. [see: 78]. Panel models were then applied on the matched dataset to estimate the effect of the variables of interest. The matching procedure was implemented as follows. First, we created a cross-sectional dataset of all properties and a treatment binary variable if the property was ever occupied or not. We then calculated the propensity score of being occupied or not through a logit regression using cost-distance to cities, cost-distance to roads, property size, amount of forests on the property, and quality of soils for agriculture. A property in one group (occupied or not) was considered to have a similar property on the other if their propensity scores were within a caliper distance of 0.063, or 0.55 standard deviations of the scores, after Austin [79]. Overall, the original sample was already balanced with the exception of one property (see S1A Fig and S1B Fig).

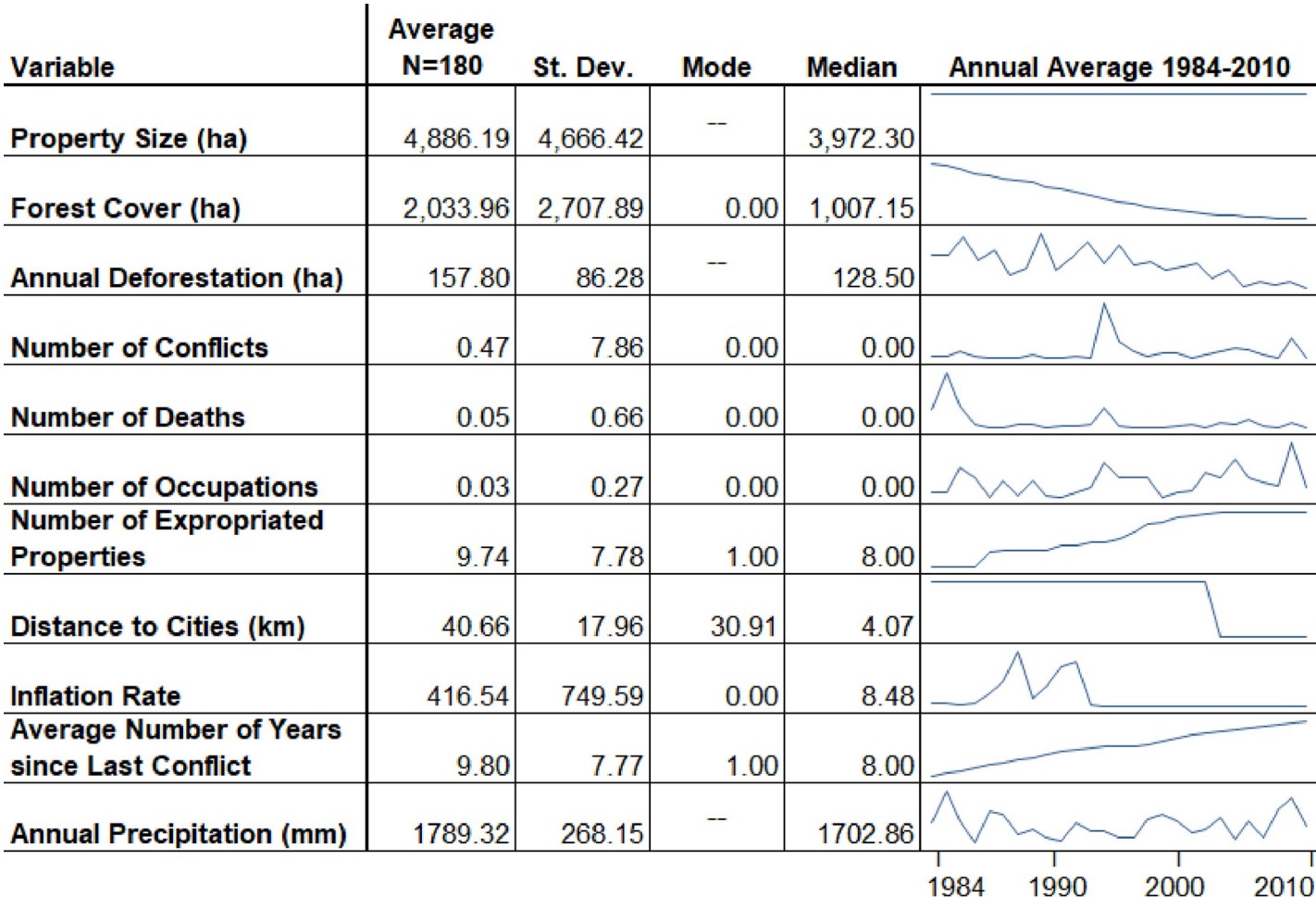

| Variable | Average N=180 | St. Dev. | Mode | Median | Annual Average 1984-2010 |
|---|---|---|---|---|---|
| Property Size (ha) | 4,886.19 | 4,666.42 | -- | 3,972.30 | |
| Forest Cover (ha) | 2,033.96 | 2,707.89 | 0.00 | 1,007.15 | |
| Annual Deforestation (ha) | 157.80 | 86.28 | -- | 128.50 | |
| Number of Conflicts | 0.47 | 7.86 | 0.00 | 0.00 | |
| Number of Deaths | 0.05 | 0.66 | 0.00 | 0.00 | |
| Number of Occupations | 0.03 | 0.27 | 0.00 | 0.00 | |
| Number of Expropriated Properties | 9.74 | 7.78 | 1.00 | 8.00 | |
| Distance to Cities (km) | 40.66 | 17.96 | 30.91 | 4.07 | |
| Inflation Rate | 416.54 | 749.59 | 0.00 | 8.48 | |
| Average Number of Years since Last Conflict | 9.80 | 7.77 | 1.00 | 8.00 | |
| Annual Precipitation (mm) | 1789.32 | 268.15 | -- | 1702.86 | |

**Fig 3. Summary statistics for important characteristics of BNP properties.**

**3.b.ii) Data.** Given the long-run specification of our hypotheses, a panel dataset including annual observations of forest cover and land contention was required. Annual land cover maps were created by classifying 52 Landsat 5 TM images (path 223, rows 64 and 65) using a hybrid classification method similar to Simmons et al. [32] and Aldrich et al. [44]. Annual contention data was collected for each property from newspaper accounts published in two regional newspapers, *O Correio do Tocantins* and *Opinião*!. These are the two newspapers from the region that reliably published semi-daily or daily newspapers over the entire study period (two other papers published intermittently, but did not provide sufficient coverage to include in the systematic dataset). Both newspapers began consistent publication in 1984. Each newspaper account was read, and salient details of the conflict recorded (e.g., number of deaths, police involvement, names of property owners, character of the conflict, what type of conflict events took place) and summarized (see S1 File for complete details). Accounts from both newspapers were then compared, and when events were reported in both newspapers, which was frequent, the details were combined into single reports for each property in the statistical dataset. We matched events to properties using the property cadastral described above [80], matching by property name, location (generally indicated by municipality—this study area covers parts of nine municipalities), and owner name (see S1 File for details). For this

analysis we constrain the time period to correspond with the redemocratization of Brazil after twenty years of a military dictatorship (i.e., 1984) through the date when the State of Pará, Brazil formally transferred the large properties of the study area into private hands in 2010 [46]. The study period we employ has other benefits, including placing a hard limit on the range of archival research in difficult conditions (including a lack of formal records on conflict in the region before approximately 1982), and the fact that the majority of properties in the region were totally, or close to totally (single-digit forest cover proportions), deforested by 2010.

A variety of control variables were used, all in relative agreement with well-established drivers of land change [e.g., 81]. Road data that show the temporal evolution of the road network are difficult to find in most cases, but data was made available for this project by Walker et al. [82] and updated to 2010 extent using roads data provided by the Brazilian Institute for Geography and Statistics. City locations were acquired from the Brazilian National Institute for Geography and Statistics. Distance to cities was generated using the roads data, represented in kilometers. Inflation Rate was acquired from the World Bank. Annual precipitation, meant to control for climate variation over the study period [one of the early observable effects in the region, see 83], was acquired from NOAA/OAR/ESRL PSD (http://www.esrl.noaa.gov/psd/). In some cases we interact or decompose explanatory variables to better understand potential drivers of change.

The data sets were integrated at the property level acquired using a cadastral map of 180 large landholdings published by the Superindendency for Amazonian Development [80], a now-defunct government agency. The paper map was digitized at 1:50,000 scale, and attributes from the map, such as property size, name, owner name, and title status (i.e., definitive title, long term lease, unknown, no title) were associated with each property and included with the above described datasets. Agrarian reform settlements and associated expropriation records were acquired from the Brazilian National Institute for Colonization and Agrarian Reform (INCRA) and were matched these to our cadastral map using overlay analysis.

Fig 3 includes summary data for most of the variables employed in this study. Average property size, 4,886.19 hectares, stays constant throughout the study period as the unit of analysis is properties and none are subdivided. Forest cover declined steadily from a substantial proportion of forest cover in 1984 to very little in 2010 (from 11.5% deforested in 1984 to 92.2% deforested in 2010). Conflict events, which we would expect to increase deforestation in most cases given a need by both largeholders and the landless to undertake production and lay claim to the land, varied, with the average of 0.47 conflicts per property per year (but with some very notable spikes in conflict, particularly in the mid-1990s and mid-2000s). Of the 180 properties in the study, 99 of them (55%) experienced some sort of conflict, and in a given year the number of land-related conflicts on contentious properties could range from 0 to 45 distinct conflict events. The number of deaths, which we include to proxy the intensity of conflict, and we expect to increase deforestation, remained low on a per-property basis throughout the study period, with the notable exception of 24 deaths on one property in the mid-1980s. On a property-level, 29.4% experienced violent conflict which resulted in a fatality, with the number of deaths ranging from 1 to as many as 24 per year. Occupations, which we expect would significantly increase deforestation, were similarly low on average, but highly variable across the study period, with notable peaks every few years throughout. On a property-level, 35% experienced at least one occupation over the 27 year period, with some properties experiencing as many as 24 occupations in a single year (Fig 3). Expropriations of properties for agrarian reform, which we also expect would increase deforestation given the need of new smallholders to establish production, start at 0, but increase steadily each year. Distance to cities, which we would expect to be associated with reduced deforestation as distance increases, remains very steady over the two and a half decades of this study, although there is a notable drop in 2003 as

the road network is developed (though this drop is an artifact of better data being available in 2003, Fig 3). The inflation rate, which we expect would be associated with increased deforestation, as land clearance can be a hedge against high inflation, is high and volatile in the pre-Real period (pre 1992), but is more stable and lower in later periods (Fig 3). Precipitation, which we expect would have a complicated relationship with deforestation varies year-to-year, with no noticeable increasing or declining trend, despite some indicate that precipitation may be decreasing in this area, probably due to periodic drought associated with climate change [83]. Low amounts of rainfall may be associated with increased deforestation, as agricultural production may decrease during dry periods. We would also expect high rainfall to be associated with decreased deforestation because logging in very wet conditions is difficult or impossible in this region. We also divided our dataset into three equal time periods of nine years (period 1: 1984–1992, period 2: 1993–2001, period 3: 2002–2010) in order to control for semi-decadal trends in OLS and pooled OLS model specifications.

## 4) Results and discussion

The results of our analysis mostly support each of our hypotheses, with deforestation being affected by certain types of conflict, deforestation being much greater on properties which were expropriated for agrarian reform settlements, and end-of-period deforestation significantly higher on properties with conflict and settlement formation. We address each hypothesis in turn, before turning to a discussion of what these results indicate for CLC and LCS more generally.

### H1: Deforestation is greater during those time periods with land conflict compared to periods with relative peace

Deforestation appears to be moderately lower during (and in the year after, see S2 Table for models with lagged variables) contentious periods which involve violent conflict (as proxied by Number of Deaths), a result that remains consistent throughout all hypothesis 1 models. In Table 1A we produce a pooled OLS regression model with detrended (first-difference) deforestation as the dependent variable. This model indicates that deforestation history is a statistically significant indicator of current deforestation, albeit with a low coefficient value. Violent land conflict is also significant, indicating that there is more than eight hectares less deforestation when deaths occur. Land expropriation and settlement formation is also significant driver of deforestation, increasing deforestation by more than 30 hectares, and the passage of time also has an effect on deforestation, with less deforestation occurring each year (due, in part, to the fact that in later periods there is less forest to cut). Properties with good soils have less deforestation (13.1 hectares). Table 1B presents an alternative specification to address Hypothesis 1 which decomposes the conflict and settlement formation variables into separate categories. In Aldrich et al. [44], whose analysis extended until only 2003, conflict was a more significant modifier of deforestation than was settlement formation, and the results of Table 1B contradict that analysis to some extent, showing that, while conflict does affect deforestation (especially in cases where settlements have been formed, at 30.6 hectares more deforestation), settlement formation is the variable which increases deforestation more substantially (including in cases where no conflict occurred at all, 24.88 hectares). Table 1B also shows that violent conflict (number of deaths) leads to slightly reduced deforestation (1.3 hectares), which, combined with the years since last conflict variable in Table 1C, suggests a wait-and-see response to conflict, which is contrary to our expectations, but has been suggested in other studies [44, 47]. Table 1A and Table 1B also indicate that property size is an important control variable, likely because largeholders eventually deforest their entire landholding if afforded the

**Table 1. H1 regression results.** See S6 Table for more details regarding Table 1C.

| Dep. Variable: | | 2a. Pooled OLS, First Difference Deforestation (Hectares), Detrended for Time | 2b. Pooled OLS, First Difference Deforestation (Hectares), Detrended for Time | 2c. System dynamic panel-data estimation, First Difference Deforestation (Hectares) |
|---|---|---|---|---|
| Regression Characteristics | | $n = 4440$, F [10, 4429] | $n = 4440$, F [11, 4428] | $n = 4261$ |
| | | Prob > F = 0.0000 | Prob > F = 0.0000 | Prob > chi2 = 0.0000 |
| | | $R^2 = 0.4513$ | $R^2 = 0.4514$ | |
| Variable Name | | Coefficient (SE) | Coefficient (SE) | Coefficient (SE) |
| Lagged Deforestation | | 0.426 (0.07)*** | 0.426 (0.06)*** | 0.399 (0.05)*** |
| Number of Conflict Events | | -0.234 (1.23) | | -0.608 (1.94) |
| Conflict Occurred (Y/N) | Settlement Formed (Y/N) | | | |
| N | Y | | 24.488 (6.47)*** | |
| Y | N | | -5.942 (8.51) | |
| Y | Y | | 30.622 (9.55)** | |
| Number of Deaths | | -8.500 (3.87)** | -8.749 (3.22)** | -12.479 (5.57)** |
| Settlement Formed | | 31.279 (6.65)*** | | 133.823 (156.03) |
| Years Since Last Conflict | | 0.134 (0.35) | 0.094 (0.45) | -15.101 (3.19)*** |
| Annual Precipitation | | -0.026 (0.01)* | -0.025 (0.01)* | -0.015 (0.02) |
| Property Size (Hectares) | | 0.016 (0.01)*** | 0.016 (0.004)*** | |
| Distance to Cities (Km) | | 0.020 (0.19) | 0.021 (0.190) | |
| Soil Binary | | -13.120 (-2.62)** | -13.703 (5.06)** | |
| Year | | -6.152 (0.77)*** | -6.154 (0.78)*** | |
| Constant | | 12337.54 (1556.26)*** | 12344.73 (1565.97)*** | 230.436 (48.53)*** |

Statistical significance indicated as follows

\* = 0.10

\*\* = 0.05

\*\*\* = 0.000. Robust Standard Error is presented.

chance. Good soils are associated with less deforestation (~13 hectares less). Table 1C presents a panel specification which largely confirms the results shown in Table 1A and 1B. These results are contradictory to those already reported in the literature in the sense that land conflict appears to be a less substantial and less-significant (in the case of conflict itself, measured by events, not significant) driver of deforestation than expropriation and settlement formation. It should be noted, though, that violent conflict significantly reduces deforestation by 12.4 hectares in this specification, which can, in cases of significant violence such as one extreme case where 24 people were killed, lead to substantially less deforestation overall. Other agronomic variables such as precipitation and soils are also significant (although the coefficient value for precipitation is quite small) and reflect general concepts of how they would modify deforestation, as discussed in the land change literature as a whole.

## H2: Among properties that have had land conflict, those properties that have been expropriated have greater deforestation

H2 considers just those properties that have had land conflict and aims to elaborate whether expropriation for agrarian reform settlement formation affects forest cover. H2 is not supported by two of three model specifications in Table 2 (and in SI-5). H2 is addressed in in the results of Table 2A through pooled OLS regression, and shows that settlement formation does

**Table 2. H2 regression results.** See S7 Table for details on Table 2B and S8 Table for details on Table 2C.

| Dep. Variable: | 3a. Pooled OLS, First Difference Deforestation (Hectares), Detrended for Time, Contentious Properties Only | 3b. System dynamic panel-data estimation, First Difference Deforestation (Hectares), Contentious Properties Only | 3c. Arellano-Bond dynamic panel-data estimation, First Difference Deforestation (Hectares), Contentious Properties Only |
|---|---|---|---|
| Regression Characteristics | $n$ = 2436, F [11, 2424] | $n$ = 2436 | $n$ = 2338 |
| | Prob > F = 0.0000 | Prob > chi2 = 0.0000 | Prob > Chi2 = 0.0000 |
| | $R^2$ = 0.4578 | | |
| **Variable Name** | **Coefficient (SE)** | **Coefficient (SE)** | **Coefficient (SE)** |
| **Lagged Deforestation** | 0.0403 (0.09)*** | 0.432 (0.05)*** | 0.382 (0.041)*** |
| **Number of Conflict Events+** | -0.292 (1.20) | 3.371 (2.87 | 5.130 (3.43) |
| **Number of Deaths +** | -9.753 (3.59)** | -20.180 (7.56)** | -12.541 (8.72) |
| **Largeholder Control+** | -13343.71 (4199.57)** | 19964.97 (5545.88)*** | -841.146 (10124.97) |
| **Largeholder Control*Year+** | 6.658 (2.10)** | -10.103 (2.80)*** | 0.395 (5.04) |
| **Settlement Formed+** | 42.346 (11.11)*** | 103.498 (91.92) | 110.714 (115.33) |
| **Years Since Last Conflict** | | -7.869 (4.27)* | -3.405 (4.21) |
| **Annual Precipitation** | -0.024 (0.21) | -0.040 (0.02)* | -0.026 (0.02)* |
| **Property Size (Hectares)** | 0.016 (0.004)*** | | |
| **Distance to Cities (Km)** | 0.067 (0.30) | | |
| **Soil Binary** | -15.844 (7.91)** | | |
| **Year** | -12.713 (2.28)*** | | -10.165 (5.64)* |
| **Constant** | 25477.53 (4578.67)*** | 338.876 (97.66)** | 20480.16 (11300.44)* |

Statistical significance indicated as follows

* = 0.10

** = 0.05

*** = 0.000. + = in the specification presented in Table 2C this variable is lagged. Robust Standard Error is presented.

significantly increase deforestation, but that result does not hold when panel specifications are applied. The pooled OLS Model (Table 2A) shows that largeholder control significantly decreases deforestation, albeit with a declining marginal effect, as indicated by the interaction of largeholder control with the passage of time. In other words, deforestation is greater in early time periods, but declines toward the end of the study period when there is very little forest left to remove. Violent conflict decreases deforestation, again reflecting a potential "wait and see" approach. Agronomic drivers are also significant, including the deforestation history of a property, as is the passage of time. This raises the question about the correlation of deforestation, conflict, and time, with deforestation events and conflict events being episodic on similar timeframes. This is the case, as indicated by S4 Table, which makes it difficult to tease apart deforestation that was undertaken as part of agronomic operations on properties and deforestation undertaken in response to conflict. This may explain why conflict events are typically not statistically significant in these analyses, but have been found to be significant in others [e.g., 32, 44, 84].

Table 2B and 2C address H2 using panel regressions. Table 2C includes some lagged independent variables. These models indicate that settlement formation is not a significant and substantial increaser of deforestation, and that deforestation is probably not significantly and substantially higher on rancher/largeholder controlled properties (again, due to the declining marginal effect). It is interesting to note that the unlagged Largeholder Control variable changes sign between Table 2B and Table 2C (where it is lagged); this is likely because a land change undertaken in one year is only visible the next given our remote sensing methods, and lagging largeholder control in the model presented in Table 2C is, in effect, a double-lag.

Overall, H2 is not supported, and expropriation for settlement formation on properties does not appear to significantly affect forest cover, a result that contradicts other observations in this region, and across the Amazon in general [e.g., 32, 34, 38, 44]. This is, at first, puzzling, particularly since Fig 4 shows this overarching trend visually, where the aggregate outcomes of conflict and deforestation processes can be observed. The trajectory of deforestation is the roughly the same for both (a cumulative growth curve) in later periods, but properties experiencing settlement formation have greater deforestation, particularly after 1996 (a year of many conflicts) and also some more severe upticks in deforestation from year to year. However, the previous observation regarding the difficulty of teasing apart processes which have similar episodic timings (i.e., that conflict, deforestation, and agronomic investment are correlated in time) is difficult in the panel specifications, and likely masks some of the effect of settlement formation.

### H3: Deforestation is greater by the end of study period on properties that have had contention compared to those without

H3 is supported, and was initially tested through Student's t-test, which shows that properties with conflict have significantly more total deforestation (801 hectares more) than those that do not (p = 0.0738, see S5 Table). However, such analyses do not consider intervening variables which are also important in the land change process, such as distance to markets, economic conditions, or other aspects of agricultural processes. Therefore, we implement a cross-sectional OLS regression (n = 179), similar to that presented in Aldrich et al. [44], albeit for the year 2010 instead of 2003.

Results germane to H3 are evident in Table 3, and show that some aspects of conflict significantly increase deforestation, a result that aligns with previous studies [e,g, 34, 44], but also that expropriation and settlement formation appears to be more significant and substantial in end-of-period deforestation in the BNP. It is interesting to note that settlements formed on poor soils have significantly more deforestation (286 hectares, Table 3A) than those formed on good soils (46 hectares of deforestation, Table 3A). Also important is that settlements on properties with a history of conflict do have more deforestation than those that do not have a history of conflict (183 hectares vs 157 hectares, Table 3B). Overall, these results support the idea of contention as a land change driver–though perhaps not as significant as the process of settlement formation–a tendency shown visually in Fig 5.

### Overall discussion

In effect, our analysis appears to demonstrate that some aspects of land conflict increase deforestation on a year-by-year basis, but in the BNP the settlement formation process is what drives deforestation more significantly than conflict itself, with the exception of violent conflict, which seems to modify deforestation trajectories (Tables 1, 2 and 3). A variety of model specifications place the practical effect of settlement formation on annual deforestation around 24 to 42 hectares per year (Table 1 and Table 2), but settlement formation does not seem to

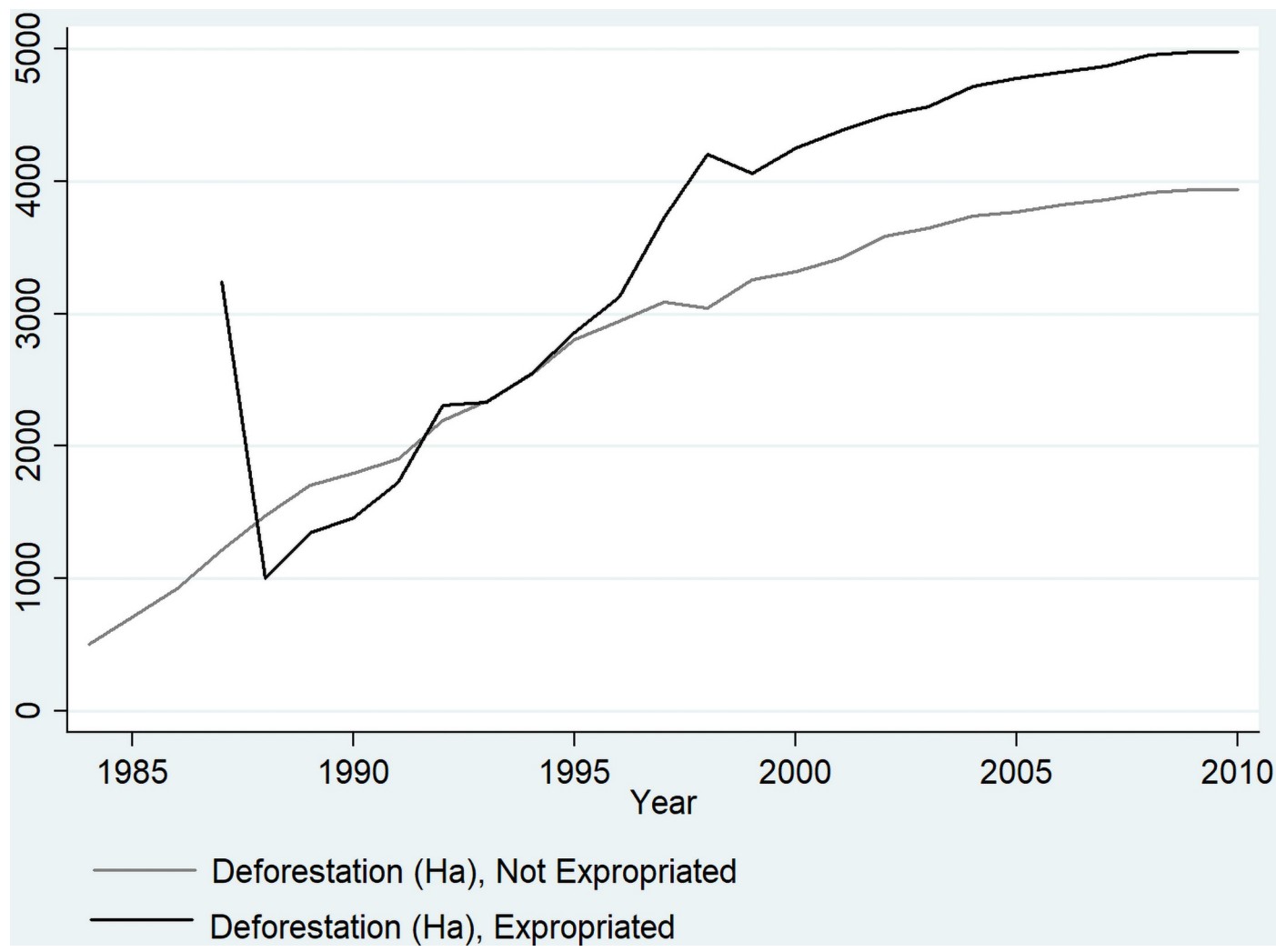

**Fig 4. Average deforested area (hectares) on expropriated for settlement formation and properties which were not expropriated for settlement formation.**

affect deforestation in panel-specifications. The fact that pooled regression specifications show settlement formation is significant, while panel-specifications do not, suggests that settlement formation occurs during a time when deforestation is also very rapid. This is supported by the line-graphs in Fig 3. When the temporal component is added through the use of panel-specifications, the results suggest that the agronomic deforestation of largeholders is neither greater nor less than the deforestation associated with settlement formation (and, in any case, these variables are not statistically significant in panel specifications presented in Table 1 and Table 2). In addition, the quality of the soil and a history of conflict both influence end-period deforestation; poor soils on agrarian reform settlements and a history of conflict on expropriated properties are both significantly associated with higher levels of deforestation (Table 3).

Although the results presented here support the idea that violent conflict which includes fatalities and the associated process of settlement formation increase deforestation, the result for the landscape of the BNP is near-total deforestation, with the 8,000 km2 of original forest reduced to less than 700 km2 by 2010. In this respect, it appears that contention may modify the process of deforestation, rather than drive it wholesale, meaning that agronomic drivers

**Table 3. H3 regression results.**

| Dep. Variable: | 4a. OLS, Deforestation (Hectares) | 4b. OLS, Deforestation (Hectares) |
|---|---|---|
| **Regression Characteristics** | $n = 179$, F [8, 170] | $n = 179$, F [7, 171] |
| | Prob > F = 0.0000 | Prob > F = 0.0000 |
| | $R^2 = 0.9805$ | $R^2 = 0.9806$ |
| **Variable Name** | **Coefficient (SE)** | **Coefficient (SE)** |
| **Number of Conflict Events** | 3.149 (14.45) | |
| **Number of Deaths** | -0.487 (3.80) | |
| **Average Precipitation** | -3.736 (2.12)* | -3.711 (2.10)* |
| **Property Size (Hectares)** | 0.951 (0.03)*** | 0.950 (0.31)*** |
| **Distance to Cities (Km)** | 4.323 (4.32)* | 4.181 (2.63) |
| **Soil Binary** | | -201.318 (118.18)* |
| **No Settlement & Good Soils** | -138.247 (129.46) | |
| **Settlement & Good Soils** | 46.333 (202.80) | |
| **Settlement & Poor Soils** | 286.405 (100.26)** | |
| **No Settlement & History of Conflict** | | -139.552 (121.70) |
| **Settlement & No History of Conflict** | | 157.316 (80.48)* |
| **Settlement & History of Conflict** | | 183.311 (102.26)* |
| **Constant** | 6250.631 (3940.85) | 6315.235 (3947.77) |

Statistical significance indicated as follows

* = 0.10

** = 0.05

*** = 0.000. Robust Standard Error is presented.

are certainly important as well. These insights lead us to revise Fig 2, to reflect the actual stylized trajectory of deforestation in the contentious agrarian landscapes of the BNP, shown in Fig 6. From our analysis it is clear that contention and settlement formation contribute to enhanced deforestation, but they by no means define deforestation in the BNP. By 2010 deforestation was greater on contentious and expropriated properties, but uncontentious properties and those that remained unexpropriated still had very little remaining forest cover.

Given what the literature on land conflict and deforestation suggests [e.g., 32, 34, 38, 44], our results are somewhat confirmatory of past suppositions and empirical observations, but our results place more emphasis on overall deforestation trajectories—agronomic deforestation happens *alongside* contentious land change. Our results may differ because we arrived at them through a 27-year panel dataset, providing a much more rigorous test of the effect of conflict on forest change than most studies have previously attempted. Furthermore, land conflicts often involve large properties, for which cadastral information is hard to come by; the analysis presented here is unique for the Amazon basin, given our access to property boundaries for a contiguous set of largeholdings. The results overall uphold the idea that agrarian struggle, and the response of the State to it, has modified deforestation in the Brazil Nut polygon in the arc of deforestation, but that violence in conflict significantly slows deforestation in almost all of our models (Table 1 and Table 2). Data limitations restrict the spatial extent of the analysis; that said, agrarian reform settlements are common throughout the basin, and land conflicts have been endemic in Brazil, as well as throughout South America. When forests stand in regions where adversarial claimants vie for the same piece of land, and agronomic land uses are growing, excessive deforestation is a likely consequence.

A few of the outcomes from these analyses are interesting given their mixed, or contrary, indications. For example, conflict events seem to decrease deforestation in some models

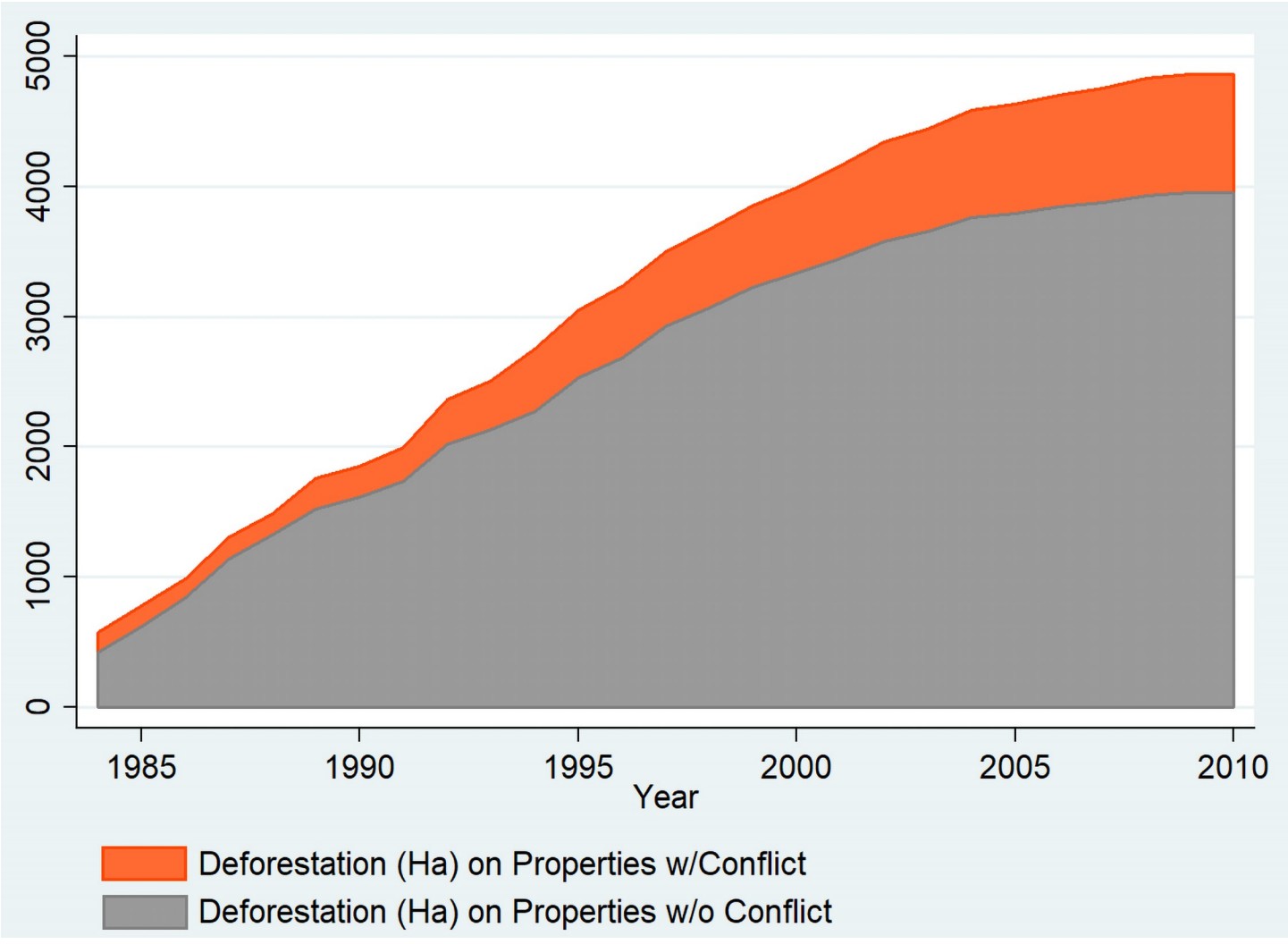

**Fig 5. Analysis results indicate that the CLC/ALC process looks more like this trajectory rather than the one presented in Fig 2.**

(Table 1A, Table 1C, Table 2A), although this is not a statistically significant result. This finding contradicts previous work in the BNP [44], and across the Amazon as a whole [84]. However, this seemingly contradictory result probably stems from the level of detail in explanatory variables in this study [i.e., we have more specific independent variables at finer spatial resolution than 8t], and employ a 27-year panel dataset [unlike 44].

One interesting outcome of this analysis is the insight that rancher/largeholder control of a property may greatly increase deforestation (Table 2B), although this increase has a declining marginal effect (which eventually becomes negative in later time periods). However, deforestation histories appear to be more significant than settlement formation, as demonstrated by the constant significance of lagged deforestation as an explanatory variable (see Table 1 and Table 2).

Another outcome that has not been investigated in detail before is the relationship between deforestation and the quality of soil for agricultural uses, particularly in cases of agrarian reform settlement formation. Our results indicate that settlements on poor soils have much higher amounts of deforestation (240 hectares more deforestation), and that deforestation is

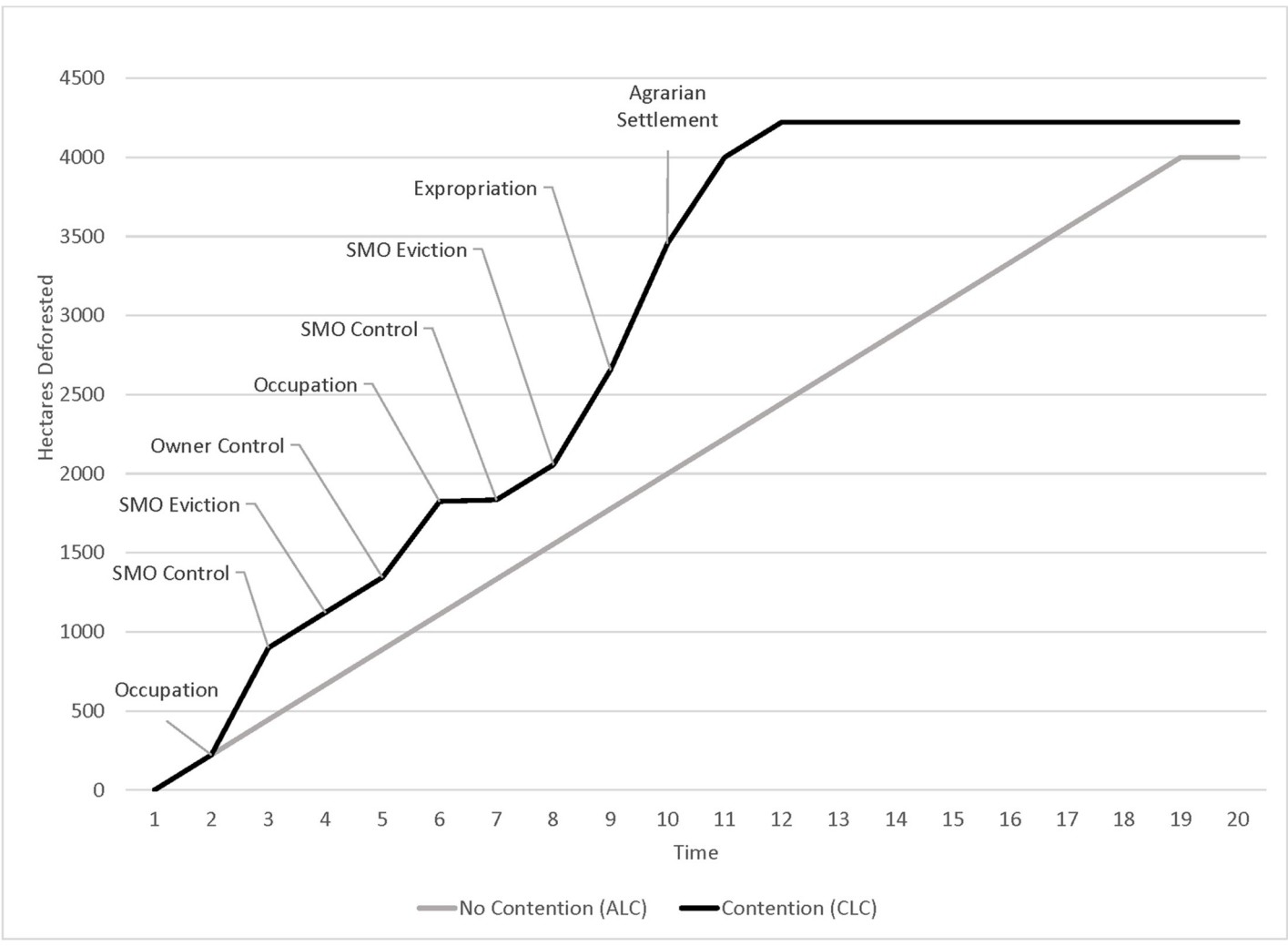

**Fig 6. Average deforestation in hectares for properties with conflict and without conflict.**

significantly lower on properties with better soils, suggesting that agrarian reform agencies may be able to help avoid deforestation through preferentially establishing settlements in areas with better soils (Table 3A). Finally, though settlement formation seems to have a more stable, significant, and substantial influence on deforestation across the 27 years of this analysis, a history of conflict on a settlement (in the time before the settlement was created) significantly increases overall deforestation by the end of the period (Table 3B).

Even though some aspects of contention drive deforestation according to our results, the Eastern Amazonian study area reveals nearly total clearance (with most remaining forest in indigenous reserves and conservation units). While some properties in the Brazil Nut polygon still retain remnants of their cover, by far most of it has been converted to fields and pastures. This is consistent with our results, given that agronomic and contentious forcing of land change has long afflicted the region, now home to a large concentration of agrarian reform settlements and neighboring large ranches. As the statistical findings show, settlement formation may lead to additional magnitudes of deforestation beyond what occurs under ALC alone, but *violent* conflict may mitigate deforestation somewhat, leading to "wait and see" responses where deforestation is avoided, perhaps given enhanced scrutiny by the public. Nevertheless,

the magnitude of deforestation attributable to settlement formation and avoided deforestation when violent conflict occurs, although significant in a statistical sense in most cases, are less important in empirical terms when considering the size of holdings in question, nearly all of which cover thousands of hectares (13 of the 179 properties are under 2000 hectares in area). This is to say that in the aggregate, ALC probably accounts for a significant amount of deforestation, with CLC modifying deforestation trajectories in times, and locales, of conflict. Moreover, the magnitude of forest loss is far in excess of what would be expected had the largeholders kept forest reserves of 50 percent, as prescribed by Brazil's forestry code. Thus, the results indicate that deforestation has been excessive, and that it stems from the region's land managers acting both independently and in competition with SMOs. This is an important finding given that many land change studies assume a unitary decision-maker who (rationally) maximizes profit. Instead, agricultural decisions may be made by multiple decision-makers, and when conflict is present the multiple decision-makers generally chose to remove more of the Amazon's forests.

Although not significant in the analyses that address Hypothesis 2 (Table 2), these results also indicate that expropriation for agrarian reform settlement formation is a significant increaser of deforestation, in both the statistical and practical sense (see Table 1, and Table 3). In many respects, land reform in this region has put forests at even higher risk, placing some substantial part of the blame for deforestation in the BNP on the project of agrarian reform. While this finding should not be discounted, all land change decisions should be evaluated in some context, and the context of already-cleared areas on expropriated large properties in the BNP is one of agricultural decimation. Indeed, newly formed agrarian settlements may have had only one option for fertile and productive household agriculture; deforestation to avoid long-ranched soils and pastures choked with invasive weeds. This is not to say that the small agriculturalists who succeeded the large ranchers on those expropriated properties have no part in the blame for the near-total deforestation of the BNP, but their part should be understood in the context of the struggle of the social movements they may have been part of. Indeed, we take criticisms of the solely statistical treatments of conflict to heart [e.g., 85], and suggest that the complex social mechanisms and history of this region, described elsewhere [e.g., 32, 41, 45, 54, 86, 87], provides ample context for these mixed results. Furthermore, the legacy of deforestation on properties, as measured by the inclusion of lagged deforestation in our models, is significant in all model specifications; regardless of conflict deforestation would have happened in this landscape, and in all likelihood the endpoint of almost total deforestation was unavoidable given the state of environmental laws and enforcement in Brazil over the time period covered by this analysis.

## 5) Conclusions

Our analysis shows that land change in Brazil, and deforestation in particular, results from both complex social processes and individual behaviors. Overall deforestation in the BNP may not be completely dominated by CLC, but CLC contributes significantly, both in statistical and practical terms, to the overall evolution of land change in the region. Deforestation is greater on the properties of the BNP due to the CLC-affiliated process of settlement formation in the time before the BNP's ultimate privatization in 2010, and settlement areas are among the most deforested. We have also shown that land conflict is not necessarily a significant driver of deforestation, which contradicts other studies. However, it is true that violent conflict that results in fatalities appears to have a diminishing effect on deforestation, reducing deforestation by approximately 12 hectares per fatality. The enhanced scrutiny over land management that comes after fatal conflicts over land is evident from newspaper accounts of conflict, and

likely translates into management decisions which avoid illegal deforestation. Overall, as other studies have indicated in more general terms, land conflict and its historical solution (i.e., settlement formation) increase deforestation as they also increase strife and violence.

The results presented here also make a compelling case for the consideration of so-called social process drivers of deforestation; as elaborated in our pooled OLS and fixed effects panel specifications, CLC is a significant modifier of deforestation in this region. We suggest that similar situations arise in other parts of the world, in which case efforts to mitigate global climate change by the carbon sequestration of standing forests must consider the circumstances that put the poor at risk to social mobilization. We also suggest that land change science must pay attention to cases where the assumption of unitary decision-makers and the primacy of economic drivers of land change may fall apart, and consider carefully how social interaction may generate unexplored drivers of change, such as CLC.

## Supporting information

**S1 Fig. Deforestation trajectory on property that was excluded from our analyses after the balancing procedure.**
(DOCX)

**S1 File. Procedures for newspaper data.**
(DOCX)

**S1 Table. Variable definition table.**
(DOCX)

**S2 Table. Lagged models for hypothesis 1.** Note that lagging does not change the sign of the coefficient for conflict and agrarian reform settlement variables.
(DOCX)

**S3 Table. t-test on deforestation totals (1984–2010, measured in hectares) on contentious properties between those which were expropriated for settlement formation and those that were not.** Among properties with conflict those which are expropriated have significantly greater deforestation than those that were not. Note that 99 of the 180 properties had land conflict over the study period.
(DOCX)

**S4 Table. Conflict and the passage of time are correlated (meaning that conflict is episodic, but concentrated in certain time periods).** Related to this, deforestation and time are correlated, too.
(DOCX)

**S5 Table. t-test on deforestation totals (1984–2010, measured in hectares) between properties with conflict and those without.** Properties without contention have significantly less deforestation than those that do, an outcome that supports H1.
(DOCX)

**S6 Table. Arellano-Bond dynamic panel-data estimation for Table 1.**
(DOCX)

**S7 Table. Arellano-Bond dynamic -panel data estimation for Table 2B.**
(DOCX)

**S8 Table. Arellano-Bond test for Table 2C.**
(DOCX)

**S1 Aldrich data.**
(ZIP)

## Acknowledgments

South American Precipitation data provided by the NOAA/OAR/ESRL PSD, Boulder, Colorado, USA, from their Web site at http://www.esrl.noaa.gov/psd/. The full dataset necessary to reproduce these findings is included in the supporting information for this manuscript and is also available at Sycamore Scholars, the institutional repository at Indiana State University, http://hdl.handle.net/10484/12380.

## Author Contributions

**Conceptualization:** Stephen P. Aldrich, Cynthia S. Simmons, Eugenio Arima, Robert T. Walker, Fernando Michelotti, Edna Castro.

**Data curation:** Stephen P. Aldrich, Fernando Michelotti, Edna Castro.

**Formal analysis:** Stephen P. Aldrich, Eugenio Arima.

**Funding acquisition:** Stephen P. Aldrich, Cynthia S. Simmons, Eugenio Arima.

**Investigation:** Stephen P. Aldrich, Cynthia S. Simmons, Eugenio Arima, Fernando Michelotti, Edna Castro.

**Methodology:** Stephen P. Aldrich, Eugenio Arima, Robert T. Walker.

**Project administration:** Stephen P. Aldrich.

**Resources:** Stephen P. Aldrich.

**Software:** Stephen P. Aldrich.

**Supervision:** Stephen P. Aldrich.

**Validation:** Eugenio Arima.

**Visualization:** Stephen P. Aldrich.

**Writing – original draft:** Stephen P. Aldrich, Cynthia S. Simmons, Robert T. Walker.

**Writing – review & editing:** Stephen P. Aldrich, Cynthia S. Simmons, Eugenio Arima.

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
