## [Decision Letter · Decision Letter 0]

19 Sep 2019

PONE-D-19-18333

Agronomic or Contentious Land Change? A Longitudinal Analysis from the Eastern Brazilian Amazon

PLOS ONE

Dear Dr. Aldrich,

Thank you for submitting your manuscript to PLOS ONE. After careful consideration, we feel that it has merit but does not fully meet PLOS ONE’s publication criteria as it currently stands. Therefore, we invite you to submit a revised version of the manuscript that addresses the points raised during the review process.

We have received two qualified reviews of the paper. Like the reviewers, I do recognize the relevance of this study and substantial efforts to compile the data used in it. The reviewers pointed out the novelty of this study and also distinct deficiencies that mostly concerns the use of clearer and more precise terms or metrics, and complementary verification of the study’s hypotheses.

We would appreciate receiving your revised manuscript by Nov 03 2019 11:59PM. To enhance the reproducibility of your results, we recommend that if applicable you deposit your laboratory protocols in protocols.io, where a protocol can be assigned its own identifier (DOI) such that it can be cited independently in the future. For instructions see: http://journals.plos.org/plosone/s/submission-guidelines#loc-laboratory-protocols

We look forward to receiving your revised manuscript.

Kind regards,

Rodolfo Nóbrega

Academic Editor

PLOS ONE

Journal Requirements:

Reviewers' comments:

Reviewer's Responses to Questions

**Comments to the Author**

1. Is the manuscript technically sound, and do the data support the conclusions?

Reviewer #1: Yes

Reviewer #2: Partly

2. Has the statistical analysis been performed appropriately and rigorously? 

Reviewer #1: N/A

Reviewer #2: No

3. Have the authors made all data underlying the findings in their manuscript fully available?

Reviewer #1: No

Reviewer #2: No

4. Is the manuscript presented in an intelligible fashion and written in standard English?

Reviewer #1: No

Reviewer #2: Yes

5. Review Comments to the Author

Reviewer #1: Relevance of the manuscript:

The overarching aim of this manuscript is to strengthen the debate on Contentious Land Change within Land Change Science by focusing on one of the globally most affected regions regarding land cover changes and social conflicts. The study focuses on a time span that includes serious political, institutional and socio-economic changes, which caused far-reaching social-ecological conflicts in the study region of Southeast Pará. The authors developed an explanatory framework that addresses the impact of individual behaviors and social processes on deforestation that expands the common knowledge gain of rational actor models. Particularly striking is the approach to combine land cover maps with contentious information obtained from regional newspapers. The results underline the importance of asymmetric power relations in negotiation processes about land use decisions, which are not only relevant for the Brazilian Amazon, but generally have a benefit for research in the Global South. The authors did an excellent job supporting each of their three hypotheses adopting OLS regression and a panel approach. In general, the manuscript addresses a little-noticed subject and contains a lot of knowledge that could turn it into a valuable contribution for the Land Change Science community. To improve the manuscript in order to meet all scientific standards some minor issues should be resolved.

Methods:

The authors refer to reports published in two regional daily newspapers (‘O Correio do Tocantins’ and ‘Opinião!’) to examine the annual contention at property level. Regarding this approach, the manuscript could be improved if further information is provided that justifies their choice of these regional newspapers. The authors should also be more explicit about the form of both data sources (e.g. printed or digital archives). The authors state that each newspaper account was read, and salient details of the conflict recorded and summarized. It is not clear in the manuscript which indicators in the text analysis were used to evaluate details as ‘salient’. It is also not clear how the allocation of newspaper reports to individual properties is done. A complete description of the text analysis should be included as supporting information.

Data presentation: Figures and Tables

The figures and tables do support the research findings. Nevertheless, the authors should take into account some specifications to increase its informative value.

Figure 1: The title should refer more strongly to the central contents of the map (e.g. BNP region, deforestation). Cartographic illustration: A color reversal would better emphasize the narrative of deforestation. The study area is not clearly recognizable neither in the main map nor in the overview map. Cartographic style elements should be clarified, e.g. by using a different line color for roads; to avoid misinterpretations based on the information provided in the legend it could be advantageous not to use colors in the overview map. The source information in the legend should be clarified and completed. The authors should pay attention to the consistent title of "Southeast of Pará".

Figure 3-4: The y-axes are not labelled.

Figure 5: I assume that the presented figure 5 does not correspond to its caption; otherwise, the authors should graphically revise the difference to figure 2.

In line 6 (p. 29) is a typo: Table 4d

Language and writing style:

In some parts of the article, the language could be adapted to make the information more precise and understandable. Words that reinforce meaning, such as "horrific" (line 5, p. 2), “unfortunately” (line 12, p. 2), “looking for fortune” (line 11, p. 7), express a subjective feeling that is scientifically incomprehensible. Instead of writing “until recently” (line 20, p. 4) or “almost complete deforestation” (line 4, p. 2), estimates should be clarified (e.g. How much forest has been cleared in the region since 1984?)

Reference list:

All references are carefully selected. They represent an essential part of the state of the art in the field of contentious land change in the Brazilian Amazon. The authors should always endeavor to refer to current reports (e.g. CPT. Conflitos no Campo Brasil, 2018). I would like to suggest additional references to the authors on the subject of land tenure and social conflicts and refer especially to current work in the field of land-use change (e.g. Baumann, Matthias) and political geography (e.g. Korf, Benedikt; Raeymaekers, Timothy).

Reviewer #2: General comment:

The manuscript contributes to the growing literature on the causes of deforestation in the Amazon. Moreover, unlike the bulk of the existing literature that tends to focus on tradition drivers of deforestation, this manuscript offers a unique insight into how land related conflicts explain observed deforestation.

Below are my specific comments:

1, It would be helpful if you add a table with variable name, definition of the variable, measurement of the variable, data source etc.

2. The first hypothesis of the paper reads “Deforestation is greater during those time periods with land conflict compared to periods with relative peace”.

Accordingly, the authors run regressions with conflict as a right hand side variable. However, conflict is a potentially endogenous variable (i.e., conflict itself could be correlated to the error term). For example, conflict could be due to deforestation. Have you tested whether conflict is an exogenous variable? If yes, it would be helpful if you present your test.

3. On page 16 of the manuscript you mentioned that you have data on title status of each property Why didn't you then include title status in your analysis? One would expect title status to predict conflict

4. The regression models presented in the manuscript do not include deforestation history of the property as an explanatory variables for current deforestation. Existing studies suggest that deforestation in the area tends to be path dependent. It would be helpful to see how robust your results are to the inclusion of deforestation history.

5. On page 12 “In the other case, the property without contention experiences ALC until 2,000 ha are cleared, or half the holding, in accordance with federal law that mandates a “Legal Reserve” of 50 percent forest …”

The empirical model does not capture the role of this law. It would be better if you could show whether there is empirical support to this. For example, testing for the coefficient of an interaction term of conflict and deforestation history (or stock forest) can be an option.

6. Table 1 presents a list of variables considered in the empirical analysis (distance to cities, inflation rate, precipitation…). It would be helpful if you could include a sentence or two for each on how you anticipate each to impact deforestation.

7. On page 18 “Distance to cities remains very steady over the two and a half decades of this study, although there is a notable drop in 2003 as the road network is developed (though this drop is an artifact of better data being available in 2003”.

This suggests clearly that there is measurement error and thus, if not addressed, biased and inconsistent coefficient estimates. Did you address this measurement error?

8. On page 14 “In order to control for potential bias arising from selective occupation, we adopted a two pronged approach whereby matching estimators were used as a pre-processing step to select a balanced sample of properties that are comparable with respect to certain pre-occupation characteristics. “

It would be helpful if the result (table/graphs) of this matching exercise are presented in the SI.

9. On page 14 “ However, many “traditional” land cover change control variables like the quality of the resource base (e.g., soil quality) and distance to market (independent of the evolving road network) are handled automatically by our panel approach [68, 69], as is spatial autocorrelation [70: 27]. “

Please clarify. When it comes to settlement and deforestation spatial autocorrelation is a serious concern (i.e., deforestation in one property tends to correlate with deforestation in neighboring properties).

6. PLOS authors have the option to publish the peer review history of their article (what does this mean?). If published, this will include your full peer review and any attached files.

Reviewer #1: Yes: Michael Klingler

Reviewer #2: No

---

## [Author Response · Author response to Decision Letter 0]

12 Nov 2019

My responses to reviewer comments are included in the cover letter.

---

## [Decision Letter · Decision Letter 1]

18 Dec 2019

Agronomic or Contentious Land Change? A Longitudinal Analysis from the Eastern Brazilian Amazon

PONE-D-19-18333R1

Dear Dr. Aldrich,

We are pleased to inform you that your manuscript has been judged scientifically suitable for publication and will be formally accepted for publication once it complies with all outstanding technical requirements.

With kind regards,

Rodolfo Nóbrega

Academic Editor

PLOS ONE

Additional Editor Comments (optional):

Reviewers' comments:

Reviewer's Responses to Questions

**Comments to the Author**

1. If the authors have adequately addressed your comments raised in a previous round of review and you feel that this manuscript is now acceptable for publication, you may indicate that here to bypass the “Comments to the Author” section, enter your conflict of interest statement in the “Confidential to Editor” section, and submit your "Accept" recommendation.

Reviewer #1: All comments have been addressed

Reviewer #2: All comments have been addressed

2. Is the manuscript technically sound, and do the data support the conclusions?

Reviewer #1: Yes

Reviewer #2: Yes

3. Has the statistical analysis been performed appropriately and rigorously? 

Reviewer #1: N/A

Reviewer #2: Yes

4. Have the authors made all data underlying the findings in their manuscript fully available?

Reviewer #1: Yes

Reviewer #2: Yes

5. Is the manuscript presented in an intelligible fashion and written in standard English?

Reviewer #1: Yes

Reviewer #2: Yes

6. Review Comments to the Author

Reviewer #1: I hereby agree to the publication of the manuscript. All comments and suggestions have been included in this review. The authors have significantly improved the quality of the manuscript. I particularly endorse the extension of the supporting material (SI-1) and the detailed description of the newspaper analysis (SI-3). For the latter I even suggest to transfer parts of SI-3 to section 3.b.ii, not least to emphasize the very detailed qualitative text analysis as an elementary part of the systematic evaluation of deforestation. I share the opinion that the high number of deaths due to land conflicts is "horrific", although I prefer the current writing style.

Reviewer #2: The authors have addressed my comments/concerns on the earlier draft. More specifically, the revised manuscript presents the empirical approach in a more intelligible way: variables are clearly defined; data have been made available; and, the justification for econometric approaches looks sound. Overall, the authors have done a great job in collecting conflict and ownership data. More importantly, however, the authors employed an empirically rigorous approach to tease out the impacts of recurring conflicts on observed deforestation in the region. The paper will be a great addition to the growing literature on deforestation in the amazon.

7. PLOS authors have the option to publish the peer review history of their article (what does this mean?). If published, this will include your full peer review and any attached files.

Reviewer #1: Yes: Michael Klingler

Reviewer #2: No

---

## [Editor Report · Acceptance letter]

30 Dec 2019

PONE-D-19-18333R1 

Agronomic or Contentious Land Change? A Longitudinal Analysis from the Eastern Brazilian Amazon 

Dear Dr. Aldrich:

I am pleased to inform you that your manuscript has been deemed suitable for publication in PLOS ONE. Congratulations! Your manuscript is now with our production department. 

With kind regards,

on behalf of

Dr. Rodolfo Nóbrega 

Academic Editor

PLOS ONE